# Saffron, Its Active Components, and Their Association with DNA and Histone Modification: A Narrative Review of Current Knowledge

**DOI:** 10.3390/nu14163317

**Published:** 2022-08-12

**Authors:** Mudasir Rashid, Hassan Brim, Hassan Ashktorab

**Affiliations:** Department of Medicine, Gastroenterology Division, Department of Pathology and Cancer Center, Howard University College of Medicine, Washington, DC 20059, USA

**Keywords:** saffron, linker histone H1, phosphorylated H3, γH2AX, HDAC1, SIRT1 and G-quadruplex, DNA, epigenetics

## Abstract

Intensive screening for better and safer medications to treat diseases such as cancer and inflammatory diseases continue, and some phytochemicals have been discovered to have anti-cancer and many therapeutical activities. Among the traditionally used spices, *Crocus sativus* (saffron) and its principal bioactive constituents have anti-inflammatory, antioxidant, and chemopreventive properties against multiple malignancies. Early reports have shown that the epigenetic profiles of healthy and tumor cells vary significantly in the context of different epigenetic factors. Multiple components, such as carotenoids as bioactive dietary phytochemicals, can directly or indirectly regulate epigenetic factors and alter gene expression profiles. Previous reports have shown the interaction between active saffron compounds with linker histone H1. Other reports have shown that high concentrations of saffron bind to the minor groove of calf thymus DNA, resulting in specific structural changes from B- to C-form of DNA. Moreover, the interaction of crocin G-quadruplex was reported. A recent in silico study has shown that residues of SIRT1 interact with saffron bio-active compounds and might enhance SIRT1 activation. Other reports have shown that the treatment of Saffron bio-active compounds increases γH2AX, decreases HDAC1 and phosphorylated histone H3 (p-H3). However, the question that still remains to be addressed how saffron triggers various epigenetic changes? Therefore, this review discusses the literature published till 2022 regarding saffron as dietary components and its impact on epigenetic mechanisms. Novel bioactive compounds such as saffron components that lead to epigenetic alterations might be a valuable strategy as an adjuvant therapeutic drug.

## 1. Introduction

Hippocrates, the “father of medicine” stated, “let food be your medicine and let medicine be your food”. Plants include a variety of bioactive components that have medicinal properties. Carotenoids are naturally occurring phytochemicals found in vegetables, fruits, herbs, and spices and are widely involved in the human diet. Several studies have demonstrated that carotenoids have anti-carcinogenic properties in two ways: they can prevent free radical chain events and interact with biological molecules [1,2]. Saffron, a spice consumed in a natural form with various food ingredients, is a rich source of carotenoids. Saffron is a stemless perennial herb that belongs to the Iridaceae family and is made from the dried stigma of the *Crocus sativus* flower. Saffron contains around 150 phytochemicals with chemopreventive, chemotherapeutic, anti-inflammatory, and antioxidant effects and has nutritional cost-effective benefits [3]. Its main components are crocins, crocetin, safranal, picrocrocin, Kaempferol, naringenin, taxifolin, lycopene, zeaxanthin, and vitamins, particularly thymine. The taste, color, and odor of saffron are due to the active compounds picrocrocin, crocins, and safranal, respectively. Apart from being used as a coloring and flavoring agent, several in vivo and in vitro studies have clearly demonstrated that saffron and its active ingredients have multiple putative biological activities, such as antioxidant, free radicals scavenging [4,5,6,7], anti-inflammatory [8,9,10], anti-allergic [11], anti-depressant [12,13] anti-arthritic [14], anti-angiogenesis [15,16,17], anti-hypertensive [18,19], anti-bacterial [20,21,22], nephroprotective [23,24,25,26], anti-aging [4], anti-genotoxic [27], anti-atherogenic [28], cardioprotective [29,30], anti-obesity [31], anti-diabetic [32,33], hepatoprotective [34,35,36,37], beneficial effects on the reproductive system [38] and anti-cancer properties [8,39,40,41,42,43,44]. Zeinali et al. reported that upon treatment with saffron, there was a decrease in serum levels of pro-inflammatory enzymes and cytokines such as cyclooxygenase-2 (COX-2), myeloperoxidase (MPO), phospholipase A2, inducible nitric oxide synthase (iNOS), proteinoids, tumor necrosis factor alpha (TNF-α), nuclear transcription factor κB (NF-κB) p65, interleukins (IL) such as IL-1β, IL-6, IL-12, IL-17A and interferon-gamma (IFN-γ) [45].

Furthermore, various studies have shown that combining saffron extracts with chemotherapy have synergistic benefits, improving the success of the treatment in lung cancer and osteosarcoma [46] by reducing DNA damage, and protecting normal cells from the genotoxicity associated with chemotherapy drugs [47]. Recently, microarray analysis of mice pre-conditioned with saffron for 5 days used (~0.25–0.33 mg/kg/day) much lower daily dose (the range used an aqueous extract of saffron (80, 160, and 320 mg/kg body wt.), crocin (100, 200 and 400 mg/kg body wt.), safranal (0.1, 0.2 and 0.4 mL/kg), were used in different studies) [48,49,50,51]. They reported that 424 differentially expressed genes, (e.g., Cyr61, Ndufs4, Gpx8, and Nos1ap) with known neuroprotective effects against Parkinson’s and Alzheimer’s disease [52]. Moreover, studies have reported that saffron aqueous extract (SFE) has antioxidant/anti-inflammatory effects via regulating the expression of NRF-2 and its downstream targets HO-1 and GPX-2 in Dextran sulfate sodium (DSS) induced colitis. Similarly, they have also, shown that in a dose-dependent manner reduced stool calprotectin levels and increased anti-inflammatory markers (IL-10, TGF-β) while decreasing pro-inflammatory cytokines (TNFα, INFγ, IL-6, IL-2, IL-17a) in a dose-dependent response of SFE (250 and 500 µg/mL) [53,54,55]. However, how saffron alters gene expression profile has not yet been investigated. Recent studies have shown that extrinsic and intrinsic environmental factors, such as pollutants, hormones, and active dietary components alter epigenetic mechanisms. Epigenetics correspond to heritable and reversible variations in gene expression patterns that do not arise from alterations in DNA sequence [56]. In the mammalian system, the main mechanisms of epigenetic regulation are DNA methylation, histones modifications [57], the replacement of canonical core histones with specialized variants, repositioning or eviction of histones from DNA leading to chromatin remodeling, and non-coding RNAs [58,59,60].

Dietary components have been shown to induce epigenetic alterations [61]. Polyphenols, carotenoids, and selenium, have an effect on DNA methylation [62,63]. Moreover, other factors include butyric acid and sulforaphane, which is abundant in cheese and broccoli, respectively, and can alter histone post-translation modifications by blocking histone deacetylases (HDACs) [64,65]. Curcumin is an antioxidant spice that leads to histones’ post-translational modifications [66]. Additionally, hyper-methylation of DNA promoters of tumor suppressor genes has also been linked to high-fat diet [67]. These dietary components can impact epigenetic mechanisms, implying that they can play a role in different cancer development and prevention [68]. Here, we aim to present a comprehensive review of up-to-date literature about the interplay between saffron, its components, and the regulation of epigenetic processes.

## 2. Interaction of Saffron with Linker Histone H1, B-DNA, and G-Quadruples DNA

About 2 m of long DNA is packaged in a micron-scale nucleus in eukaryotes as a fibrous architecture known as chromatin. Chromatin is a nucleoprotein complex consisting of DNA, core histones (H2A, H2B, H3, H4, and H1), and non-histone proteins (lamins, topoisomerase II, scaffold proteins). The linker histone (H1) binds entry/exit site DNA in the nucleosome to form a chromatosome [69,70,71,72] and further strengthens a more condensed chromatin state which is closed and is transcriptionally “inaccessible” for trans-acting factors resulting in the regulation of gene expression. Furthermore, it is known that H1 dissociation from DNA causes transcriptional activation of multiple genes. Ashrafi et al. have reported the interaction between Saffron active compounds (like Crocetin, crocin, and dimethyl crocetin) with linker histone H1. The binding interaction was further assessed, and it was found that Crocetin and dimethyl crocetin interact more strongly than crocin with the histone H1 [73] (Figure 1A). Based on the results, the Saffron active compounds strongly bind to linker histone H1. Hence, they have the capacity to alter its conformation and affect its interaction with DNA, thus exerting their anti-cancer activity by depleting linker H1 from linker DNA and promoting selective transcription of genes associated with cell survival by some mechanism that needs to be further studied. Additionally, it is essential to understand which H1 variant or isoform is affected by saffron. Generally, H1 acts as a repressor; however, there are few reports that H1.C and H1.E function as activators. Saffron constituents’ effect on H1 histone isoforms needs to be further deciphered to assess which specific isoforms are affected by each saffron compound and as a result change in gene expression (upregulated or downregulated genes set).

Hoshyar et al. have shown that saffron carotenoids (Safranal and picrocrocin) bind to the minor groove of calf thymus DNA and synthetic oligonucleotides to induce specific structural changes from B- to C-DNA form at high concentration of saffron [74,75] (Figure 1B). Protein–DNA complexes have been found to reveal that an A-like DNA conformation can form when specific proteins bind to canonical form (B-DNA) or can be an essential intermediate step in producing the severely distorted DNA conformation seen in at least some protein–DNA complexes [76,77,78].

Studies have shown that telomeres of eukaryotic chromosomes play a vital role in cell survival. The G-quadruplex and I-motif are two distinct non-canonical secondary structures in eukaryotic chromosomes, such as telomeric DNA. Binding to specific proteins or small chemicals can stabilize or modify these structures. Hoshyar et al. have shown that crocin interacts with G-quadruplex [79] (Figure 1C), suggesting this interaction could be one of saffron’s anti-cancer mechanisms. However, the consequences of this interaction might trigger various epigenetic changes that need to be further elucidated.

## 3. Saffron Modulates the Expression Levels of Histone Deacetylases, Phosphorylated H3, and γH2AX

The mammalian sirtuin families (SIRT1-7) are made up of seven conserved proteins that participate in a range of cellular functions. Sirtuin 1 (SIRT1) is a nicotinamide adenine dinucleotide (NAD)+-dependent protein deacetylases that acts as a sensor that directly connects metabolic perturbations with transcriptional outputs [80]. Various reports have shown that SIRT1 is not just a histone deacetylase; it also interacts and regulates the activity of many co-regulators and transcription factors such as FOXO [81], p53 [82], PPARγ [83] and regulates various biological functions. Previous reports have shown that activating Sirtuin 1 increases the AMP-activated protein Ser/Thr kinase (AMPK), Nrf1/2 [51], and PGC1 alpha expression [84,85]. Other reports have demonstrated that AMPK phosphorylates Nrf2 at serine 558 residue favor its nuclear translocation, resulting in an enhanced endogenous antioxidant system and mitochondrial biogenesis [86,87].

Recent studies have shown that ligand interaction and mutation analysis of SIRT1 (Ile223, Ile227, and Asn 226 residues) were responsible for enhanced SIRT1 activation [88]. Similarly, another study has shown that Resveratrol interacts with Asn226 residue of SIRT1 and stimulates its activity [89]. Interestingly, Saffron bio-active compounds (Crocetin, Safranal, and Picrocrocin) interact with Asn226, Arg1, Ile223, Ile227, and Lys3 residue of SIRT1 via hydrogen and hydrophobic interactions with different degrees; however, other residues of SIRT1 showed weak interaction with saffron. Hence, considering the ligand binding energies and strength of hydrogen bonding between SIRT1 and saffron bio-active compounds, Crocetin can be a better allosteric activator of SIRT 1 than Resveratrol [90], as shown in (Figure 2. Furthermore, other studies have shown increased expression of SIRT1 transcripts in mice breast tumor tissue after 4 weeks of HIIT (high-intensity interval training) and aqueous saffron extract mice group compared to the control group, resulting in regression of the tumor growth by an unknown mechanism [91]. As such, it is essential to understand how the saffron-SIRT1 complex affects the cell’s downstream processes, which further needs to be investigated.

Several lines of evidence reported that the Topoisomerase I (TOP1) enzyme catalyzes the transient breaking and rejoining of one strand of the DNA duplex by relieving supercoiling and tension ahead of the replication fork during DNA replication [92]. However, if TOP1 expression is persistently expressed in the cell, it induces DNA damage. Interestingly, cancer cells have strong DNA repair mechanisms, which provides a chance to treat cancer cells by DNA damaging agents. Therefore, tyrosyl-DNA phosphodiesterase (TDP1), a multiprotein complex (PAPR, HDAC1, and HDAC2), is usually needed to remove TOP1–DNA cleavage complexes, thus protect against DNA strand breaks due to the TOP1 deregulation. Reports have shown that the treatment by Saffron bio-active compounds, especially Safranal, increases TOP1, phosphorylates histone H2AX, decreases HDAC1 expression, and phosphorylates histone H3, and TDP1 in a time-dependent manner resulting in enhanced DNA damage in HCC cell lines (HepG2) [93]. Other reports have shown that the expression levels of HDAC1 and HDAC3 decreased in glioblastoma cell lines upon treatment by Crocetin in a dose-dependent manner (250 and 500 μM) for 72 h compared to untreated controls [94] (Figure 2). According to other studies, Safranal treatment elevated phosphorylation of histone H2AX (a DSB marker), whereas overall H2A expression remained unaltered in a time-dependent manner [95,96] (Figure 2). Intriguingly, the increased levels of p-H2AX were accompanied by a decrease in TDP1 levels. Safranal also caused downregulation of Cyclin B1, Cdc25, and histone-H3 phosphorylation resulting in cell cycle arrest. However, no direct link between saffron supplementation and dysregulation of TOP1, HDAC1, γH2AX, and pan-H3 phosphorylation (p-H3) has been demonstrated, which can be investigated further using in vitro and in vivo studies.

## 4. Discussion

We have conducted a comprehensive literature search regarding epigenetic factors and saffron from different databases including PubMed, Embase, etc., till the first of September, 2021. Saffron as a natural phytochemical has been investigated as an attractive chemopreventive agent including anti-inflammatory and anti-carcinogenic agents for multiple cancers [41,42,43]. The mechanisms underlying the cancer chemopreventive activity of saffron has being explored in depth. Despite several lines of growing evidence that saffron could be an effective adjuvant cancer therapeutic drug, researchers need to decipher the mechanisms of how saffron regulates epigenetics factors and of its anti-cancer properties. Studies have shown that trans-acting factors resulting in the regulation of gene expression such as linker histone H1 is known to be associated with a condensed chromatin state, which is closed and transcriptionally inactive. It is well established that H1 dissociation from DNA causes transcriptional activation of multiple genes. Ashrafi et al. have shown that interaction between Saffron active compounds (like Crocetin, crocin, and dimethyl crocetin) with linker histone H1 with undetermined biological consequences.

Other studies have demonstrated that minor groove-binding ligands such as mithramycin, chromomycin, polyamides, netropsin, and Hoechst are of great interest in medicine as anti-tumor, anti-microbial, and anti-viral agents [97]. Transcriptomic analysis has shown that linker histone genes were deregulated upon treatment by Safranal in HCC [93]. However, the role of saffron in regulating linker histone expression has not yet been studied. Studies have shown that saffron interacts with the minor groove of the B-form and induces conformational changes to the C-form of DNA [74]. In normal cells, telomeres protect genomic integrity, and their gradual shortening throughout cell divisions causes chromosomal instability. Telomeres length is maintained by telomerase in the vast majority of cancer cells. The length of telomeres and telomerase activity are thus essential for tumor initiation and survival [98]. Telomeric DNA has been linked to the formation of secondary structures such as G-quadruplex (G4) DNA, T-loops, and D-loops. Binding to specific proteins or small chemicals can stabilize or modify these structures. Hoshyar et al. have shown that crocin interacts with G-quadruplex [79], suggesting this interaction might trigger various epigenetic changes that need further study. Saffron and its ingredients are also being studied in preclinical settings with concurrent radiation and chemotherapy in glioblastoma and liver metastasis [99,100,101]. Saffron is a natural phytochemical with low toxicity and reduces the side effects of traditional chemotherapy medications, such as cisplatin [46], tamoxifen [102], and doxorubicin [103].

Many hypotheses have been proposed to explain saffron anti-tumor effects to be due to inhibition of various biomolecules, such as inhibiting DNA and RNA synthesis [104,105], inhibition of cancer cell proliferation [106,107], induction of apoptosis [107,108], prevention of metastasis, and angiogenesis [15,16,17,106], and alteration of tumor-suppressive genes or oncogenes expression pattern [42]. Regardless of the extensive studies, however, understanding the effect and mechanism(s) of action of saffron and its bioactive components in preventing disease progression without affecting normal cells [109] has not yet been well established. Previous studies have demonstrated that the epigenetic profiles between healthy and tumor cells vary drastically. Thus, several environmental factors (bioactive dietary components) can modulate epigenetic mechanisms and mediate and regulate gene expression [110] such as Sirtuin 1, a class III deacetylase, interacts with and regulates the activity of many co-regulators and transcription factors such as FOXO, p53, PPARγ, and AMPK. Deregulation of which Sirtuin 1 results in diverse biological consequences including neurodegeneration, inflammation, age-related disorders, heart diseases, obesity, and cancer [81,82,83,111]. Additionally, it has been shown that Isoleucine and asparagine (Ile 223, Ile 227, and Asn 226) residues of SIRT1 were responsible for SIRT1 activation [88]. Recently, in silico molecular simulation studies have shown that saffron bio-active compounds interact with these residues of SIRT1 [90]; however, the biological implications have yet to be investigated. Hence, it is important to comprehend how the saffron-SIRT 1 complex affects the downstream processes of the cell, which needs to be investigated in the future. Moreover, previous reports have demonstrated that SIRT1 can deacetylate histone H1K26, H4K16, H3K9, and H3K14 residues, resulting in the formation of facultative heterochromatin [112,113,114]. Additionally, SIRT1 has been shown to have inhibitory [115,116,117] or stimulatory [118] effects on inflammation via reduced TLR4 expression, which inhibited NFκB subunit and p65 phosphorylation and reduces ROS generation upon overexpression of SIRT1 [116].

Furthermore, high throughput RNA sequencing analysis has shown that histone deacetylases (HDAC 1, 5, 6, and SIRT 2,4,5) were downregulated upon treatment with Safranal in HCC [93]. Bajbouj et al. have studied the effect of saffron extract (0.25–5 mg/mL) significantly increased DNA double-strand breaks sensor γH2AX in HCT116p53−/− compared to HCT116p53+/+ wild-type colorectal cell lines [119], suggested that saffron induce DNA damage in a p53 dependent manner. The failure to repair DNA lesions over time has been shown to disrupt replication and transcription, leading to apoptosis [108]. In parallel, other reports have shown that saffron treatment and its active derivatives, especially Safranal, increase topoisomerase I (TOP1), and γH2AX, while total H2A expression remained unaltered. In contrast, levels of HDAC1, Pan- H3 phosphorylation, (p-H3), and tyrosyl-DNA phosphodiesterase (TDP1) were decreased in a time-dependent manner resulting in enhanced DNA damage in HCC cell lines (HepG2) [93,94,95,96], suggesting that a decrease in TDP1 levels accompanied high levels of γH2AX. Safranal also caused downregulation of Cyclin B1 and Cdc25, which resulted in cell cycle arrest and lower levels of p-H3. Furthermore, Safranal revealed a binding profile in which the aldehyde carbonyl group formed a strong H-bond with CDC25B catalytic Arg-482, suggesting a direct interaction between Safranal and CDC25B [93]. However, the link between saffron supplementation as an environmental factor and the interplay between genetic and epigenetic factors needs to be further investigated using in vitro and in vivo model systems (Figure 3).

## 5. Conclusions

The oxidant–antioxidant balance is crucial in the human body because it ensures the integrity and functionality of cell membranes, proteins, and nucleic acids. Studies on how to understand and capitalize on the antioxidant effects of several herbs and spices are underway. Saffron and its constituents exhibit high anticarcinogenic properties. Combining saffron extracts with chemotherapy has synergistic benefits without having a cytotoxic effect on healthy cells. Therefore, it is essential to determine how saffron intake is safe for the general public and how it modulates the epigenetic landscape resulting in its cellular protective effects that need to be deciphered in the future.

## Figures and Tables

**Figure 1 nutrients-14-03317-f001:**
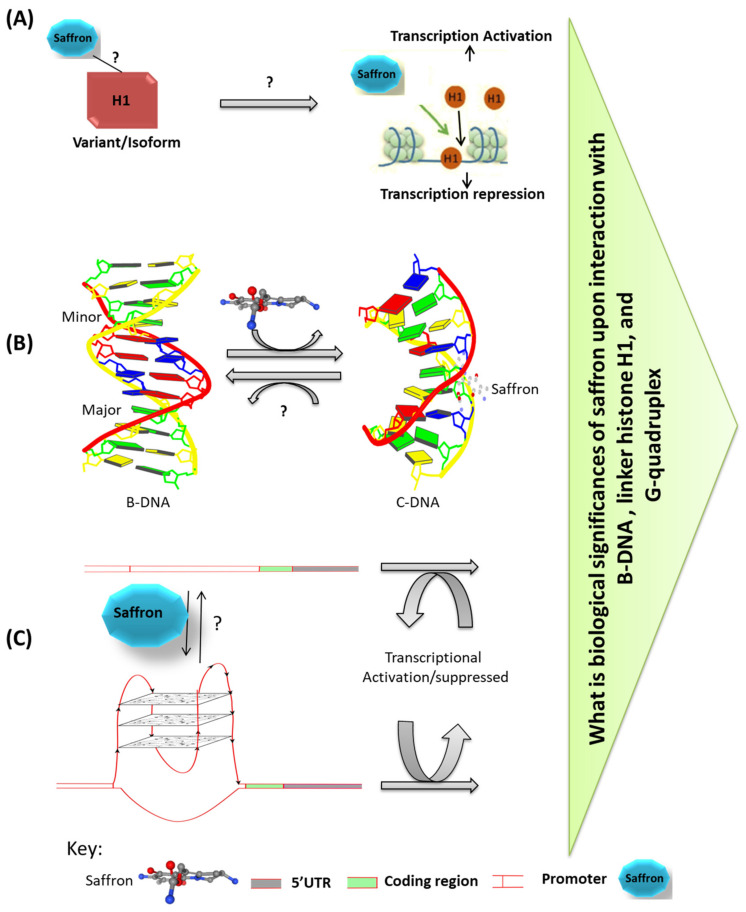
Interplay of saffron with different biomolecules. (**A**) Interaction of saffron with linker histone H1. (**B**) Saffron binds to double-stranded B-DNA minor groove and converts to C-DNA conformations. (**C**) Saffron binds and stabilizes the G-quadruples structure of DNA. The biological consequences of the interplay of saffron with H1, B-DNA, and G-quadruples have not been studied, A–blue, C–yellow, G–green, T–red.

**Figure 2 nutrients-14-03317-f002:**
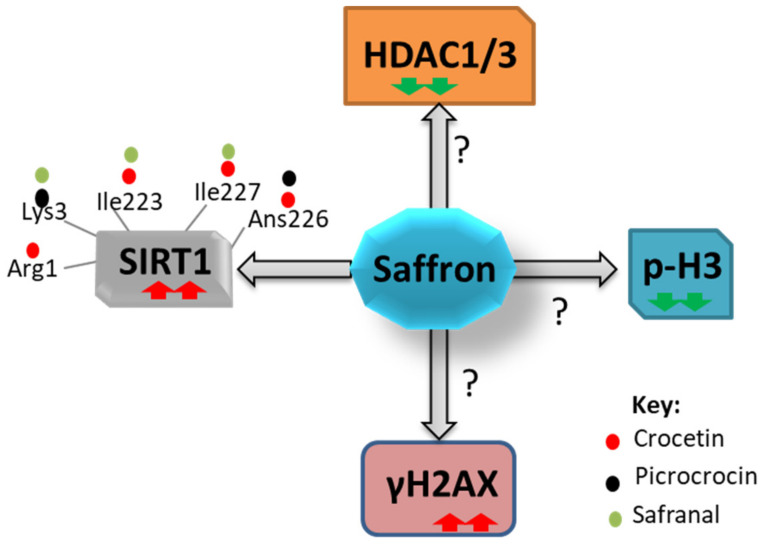
Modulation of epigenetic factors upon treatment with saffron. Saffron interacts with SIRT1 at different residues and augments its activity; also, saffron alters the expression of HDAC1/3, pan-H3 phosphorylation (p-H3), and γH2AX by an unknown mechanism. Red arrow depicts high, and the green represents low levels of expression.

**Figure 3 nutrients-14-03317-f003:**
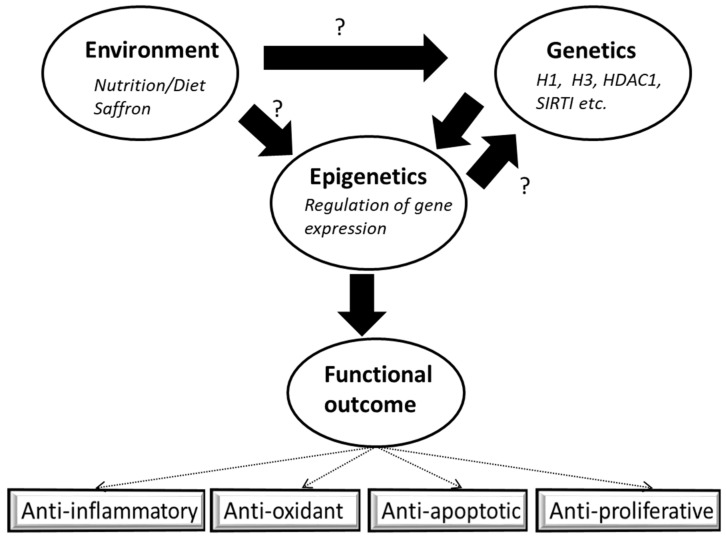
Interplay of environmental factors with genetics and epigenetics. Interaction of dietary components such as saffron and its bioactive ingredients impact several genes directly or indirectly via epigenetic mechanisms, resulting in many outcomes. How saffron alters the epigenomic landscape is still a big puzzle that needs further study in the future.

## Data Availability

No data was reported.

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
