# Peer review of "Saffron, Its Active Components, and Their Association with DNA and Histone Modification: A Narrative Review of Current Knowledge"

_nutrients, 2022, doi:10.3390/nu14163317_

Round 1

Reviewer 1 Report

An interesting approach for the impact of saffron as a chemopreventive agent.

The method of literature research should be provided along with the databases searched.

Author Response

  1. The method of literature research should be provided along with the databases searched. 

Response: Thank you for your suggestion. We have added literature and database information on saffron and epigenetics has been included in lines 204-206

Reviewer 2 Report

This is an interesting review of the potential impact that saffron and its metabolites has on epigenetic processes. What was missing in this review was a discussion of the concentration of saffron and its metabolites in foods. This would answer the question whether the doses used in studies with saffron and its metabolites are any thing like the doses of saffron added to food.

Other points to address.

When reviewing literature on the application of saffron, it is important to list the dose and time that saffron was applied. For example, the text on line 62 “ microarray analysis of mice 61 pre-conditioned with saffron for 5 days” should provide information about the dose. Same applies to line 65 for the human clinical trial. Also for line 167.

Line 65, what is DSS-induced colitis? DSS is not defined.

Line 79, rich in broccoli sprouts not so much in broccoli.

The text on line 151 needs to be edited “(REF.. Ashktorab et al, I don’t understand which reference)”

Line 171, define TOP1 when it is first mentioned in text. Same with TDP1.

Line 191, pho-H3? On line 189, is this phosphorylation of H3 at S10, S28 or another site?

The Discussion needs some revision not to re-state what was already written in the Introduction.

Author Response

Reviewer 2 

This is an interesting review of the potential impact that saffron and its metabolites has on epigenetic processes. What was missing in this review was a discussion of the concentration of saffron and its metabolites in foods. This would answer the question whether the doses used in studies with saffron and its metabolites are anything like the doses of saffron added to food. 

Response: Thank you for this valuable feedback. The range of the saffron dose were used by different groups has been included in the manuscript in lines 60-69.

Other points to address. 

When reviewing literature on the application of saffron, it is important to list the dose and time that saffron was applied. For example, the text on line 62 “ microarray analysis of mice 61 pre-conditioned with saffron for 5 days” should provide information about the dose. Same applies to line 65 for the human clinical trial. Also for line 167. 

Response: Thank you for your comments, In lines 60 and 69, the dose of saffron has been included in the manuscript

Line 65, what is DSS-induced colitis? DSS is not defined. 

Line 79, rich in broccoli sprouts not so much in broccoli. 

The text on line 151 needs to be edited “(REF.. Ashktorab et al, I don’t understand which reference)” 

Line 171, define TOP1 when it is first mentioned in text. Same with TDP1.

Response: Thank you for these valuable comments. The full form of DSS, TOP1, and TDP1has been added, also the broccoli part and reference issue (Ashktorab et al) has also been reframed and repositioned in the manuscript (line 75-77 and ref. 44 and 49 respectively).

Line 191, pho-H3? On line 189, is this phosphorylation of H3 at S10, S28 or another site?

Response: Thank you for this comment. It is a pan H3 phosphorylation (means it is a total phosphorylation level of H3) not S10, S28 phosphorylation of H3, It will be interesting to investigate which H3 residue (S10, S28) undergo phosphorylation in presence of saffron and its bio-active compounds and biological importance in terms of epigenetics.

The Discussion needs some revision not to re-state what was already written in the Introduction. 

Response: We thank the reviewer for pointing this out. Accordingly, the line of the discussion has been revised
